# Development and Feasibility Test of a Theory- and Evidence-Based Multicomponent Intervention to Reduce Student Smoking at Danish Vocational Schools

Susan Andersen [1,*], Ditte H. Holt [1], Johan L. Vinther [2], Dina Danielsen [1], Gitte S. Jakobsen [1], Teresa Holmberg [3], Marie P. Jensen [1], Charlotta Pisinger [2,4,5] and Rikke F. Krølner [1]

1 National Institute of Public Health, University of Southern Denmark, Studiestræde 6, 1455 Copenhagen, Denmark; mapj@sdu.dk (M.P.J.)
2 Department of Public Health, University of Copenhagen, Øster Farimagsgade 5, 1353 Copenhagen, Denmark
3 Center for Childhood Health, Islands Brygge 43, 2300 Copenhagen, Denmark; tho@cslt.dk
4 Center for Clinical Research and Prevention, Capital Region of Denmark, Bispebjerg-Frederiksberg University Hospital, Nordre Fasanvej 57, 2000 Frederiksberg, Denmark
5 Danish Heart Foundation, Vognmagergade 7, 1120 Copenhagen, Denmark
* Correspondence: suan@sdu.dk

**Abstract:** The smoking prevalence among vocational education and training (VET) students is high. This paper describes the development and feasibility test of a multicomponent intervention designed to promote non-smoking behaviour at VET schools. We applied the Behaviour Change Wheel (BCW) stages and the capability, opportunity, and motivation for behaviour (COM-B) model to develop the intervention components based on theory, evidence, and a thorough needs-assessment study. Moreover, we feasibility-tested the preliminary ideas. All in all, our study was based on the literature, survey data, fieldwork, workshops, and expert and stakeholder involvement. The final intervention programme targets VET students' smoking behaviour at the school, class, and individual levels through (a) a comprehensive tobacco policy to limit the physical and social opportunities to smoke, supplemented with a two-day staff course in conversations about smoking and an edutainment session (a lecture that both educates and entertains) to support a smoke-free environment; (b) classroom curricular with teaching materials to increase knowledge and social support, along with a quit-and-win competition to increase motivation; and (c) the national Quitline adapted to VET students to increase access to cessation support. The BCW model enabled a systematic and comprehensive development of an intervention, which demonstrates relevant techniques and delivery options to have the potential to reduce smoking in VET schools.

**Keywords:** behaviour change wheel; intervention development; multicomponent; youth smoking; vocational education and training schools; Denmark

## 1. Introduction

Tobacco smoking is a major public health issue and remains one of the most important preventable causes of death and diseases worldwide [1]. Adolescence and young adulthood are critical periods for tobacco prevention as they represent a time for smoking initiation and progression to regular smoking, which is strongly predictive of smoking in adulthood [2,3]. Moreover, these are important periods to target smoking behaviour because young people are particularly susceptible to nicotine addiction [4]. Despite the overall decline in smoking prevalence in Denmark and other European countries [5], there is still a high smoking prevalence in some youth populations. In particular, the prevalence of smoking among students attending vocational education and training (VET) is alarming with some countries reporting a smoking prevalence of up to 70% [6]. In Denmark, the prevalence of students smoking daily at VET schools (29%) is considerably higher than that among students

enrolled in general upper secondary education, i.e., high schools (9%) [7,8]. The student population at VET schools is characterized by a lower socioeconomic status (SES) compared with high schools. For example, 17% of Danish VET students have parents with only compulsory school as the highest level of education compared to about 4% of high school students [9]. Students at VET schools are thus an important group to target in interventions to prevent youth smoking and address social disparity in tobacco use and health [10].

The high prevalence of daily smoking among VET students points to the importance of developing interventions to assist young people to quit smoking as well as to prevent smoking initiation and escalation [11]. A meta-analysis of nine randomised controlled trials showed that targeted behavioural interventions (e.g., counselling) are successful in preventing smoking and assisting with cessation [12]. However, recruiting and retaining young people for smoking cessation interventions are a great challenge. This may be due to young people not necessarily identifying themselves as a 'smoker', but rather as an individual who smokes for a period [11,13]. Another important barrier to quit smoking among adolescents and young adults is the social influence [14]. Youth represents the period of social transition to adulthood, where individuals develop their identity and personality, accompanied by engagement with peers in new, social communities [2,15]. Accordingly, cigarette smoking within the VET school context may play a central role in socialising and facilitating new relationships [13,16,17].

One way to intervene against and decrease smoking in the school context is by implementing school tobacco policies. At the time of the current study, most VET schools in Denmark had implemented school tobacco policies, but there was a large variation in the comprehensiveness of these policies as well as their enforcement [18]. The national legislation prohibited indoor smoking. Moreover, outdoor smoking was not allowed if the school premises included both high school and vocational education, but otherwise, smoking was allowed on the school premises in designated outdoor smoking areas [19]. Tobacco policies in schools have not been studied to the same extent as smoking cessation interventions, and studies have mainly focused on students at the primary or lower secondary school level. The evidence of school tobacco policies at the upper secondary school level is therefore limited. A review of 31 studies found insufficient evidence to support that smoking policies in schools prevent smoking [20]. Nevertheless, the authors highlighted that several elements of school tobacco policies are promising, including comprehensive and clear smoking bans that apply to everyone, consistent enforcement of such bans among both students and school staff, and the presence of education and prevention programmes. An evaluation of a Danish settings-based prevention programme 'Shaping the Social' in VET schools found that fewer students progressed from occasional to daily smoking, but no effects on smoking cessation were found [16]. A conclusion was that preventive initiatives should also include individualized efforts. According to existing evidence, the most promising interventions against smoking target less advantaged communities [21] and address both individual and contextual determinants through multicomponent programmes [20,22].

However, despite the emerging literature, the evidence of effective intervention programmes to prevent the chain of events that lead towards persistent tobacco use remains scarce [11]. Particularly, programmes supported by theoretical frameworks and understanding are essential to elucidate why and how different elements or components of an intervention contribute to a potential overall effectiveness. Guidelines and frameworks for designing complex interventions that contain several interacting components are made available by the Medical Research Council (MRC) [23]. It has not been proven that one approach is better than another in the development of successful interventions [24]. To design the intervention, we decided to follow the systematic process outlined by the BCW approach [25]. The BCW integrates behaviour theories by synthesising several behavioural change frameworks in three layers. The centre of the BCW focuses on how capability, opportunity, and motivation influence change of behaviour (COM-B), which is based on the principle that behaviour change depends on the individuals' capability, opportunity, and motivation to act. The Theoretical Domains Framework (TDF) is a further

subdivision of COM-B and has 14 key theoretical constructs that link with the COM-B domains. The TDF explains what drives behaviour (e.g., skills, beliefs about capabilities, emotion, and social influences) and, thus, provides a more detailed understanding of the COM-B components [26,27]. We used the COM-B analysis to identify potential barriers to reducing smoking in VET schools, and the TDF to identify influences on peoples' capability, opportunity, and motivation. The COM-B model is encircled by nine intervention functions (education, persuasion, incentivisation, coercion, training, restriction, environmental restructuring, modelling, and enablement) and seven policy categories (communication, guidelines, fiscal, regulation, legislation, environmental/social planning, and service provision). This provides a stepwise approach moving from what needs to change to selecting specific intervention components. In addition to the BCW, we were also inspired by self-determination theory (SDT) [28,29]. A recent meta-analysis showed that SDT-informed interventions could change health behaviours, and the authors concluded that these changes were explained by changes in autonomous motivation and perceptions of the need for support [30]. The SDT highlights that developing a new health behaviour requires the individual to endorse the value of the new behaviour and develop the requisite skills for change. For example, individuals are said to be autonomously self-regulated and have intrinsic motivation if they attempt to quit smoking because it is personally important to them or congruent with their values [31]. In contrast, individuals who smoke are said to be controlled by extrinsic motivation if they attempt to stop because of peer pressure or due to internalized guilt or shame [31].

Following the recommendations by the MRC [23,32], we used a theoretical framework, based on the Behaviour Change Wheel (BCW) model, to develop a complex intervention [32–34]. In this article, we describe how we applied the BCW to develop a complex, multicomponent intervention to reduce smoking among students in Danish VET schools, which included feasibility testing.

## 2. Materials and Methods

### 2.1. Study Setting

The intervention targets 16–25-year-old young people at the basic course of VET schools across Denmark. The Danish education system includes nine years of compulsory school (corresponding to the age from 6 to 15 years), after which upper secondary school follows. The upper secondary education is divided into two separate tracks, where high schools provide general education and VET schools provide students with skills and knowledge for specific professions. While high schools have a strong academic focus and prepare students to continue into higher education, VET leads to a VET qualification and is often chosen by students who prefer non-academic learning [35]. The Danish VET system offers more than one hundred different types of vocational education directed towards skilled professions, and the prescribed duration is at least three years. The VET education is divided into four main subject areas: Care, Health and Pedagogy (CHP); Administration, Commerce and Business Service (ACB); Food, Agriculture and Hospitality (FAH), and Technology, Construction and Transportation (TCT). The VET programme includes a basic programme, in two parts, followed by a main programme. Each part of the basic program has a duration of 20 weeks. Students enrolled in the first part of the basic programme have left compulsory school within the past two years. The second part of the basic programme is an introduction to the main programme and includes students primarily aged 18–25 years.

### 2.2. Sources of Data

Our intervention design follows the steps provided by the BCW and was, moreover, guided by findings from several data sources. These included existing survey data [7–9] as well as fieldwork, workshops, and the literature, as outlined below. Moreover, the development of the intervention was based on knowledge from the Shaping the Social intervention in Danish VET schools, which included qualitative findings from managers, teachers, and students [17,36] and evaluation of effectiveness [16].

### 2.2.1. Qualitative Data from Fieldwork in VET Schools

We used qualitative research methods to explore and identify what needed to change (BCW stage 1). From January 2017 to June 2017, we conducted observations at four VET schools supplemented with focus group interviews with students (*n* = 20) and semi-structured interviews with school manager (*n* = 1), student advisor (*n* = 1), andteachers (*n* = 4). The four participating schools all offered education within either CPH or TCT subject areas. In addition, we continuously involved a VET teacher/consultant from one of the largest VET schools in Denmark, located on several campuses in Copenhagen (the capital of Denmark). A total of sixteen days of participant observations [37,38] were conducted by one researcher to examine the daily life of the VET schools and identify enablers and barriers to reduced school smoking. Observation notes were written during observations and supplemented and structured the same evening. The field notes were discussed with other members of the research team following the observations to begin formal analysis. Focus group interviews [39,40] were conducted to explore VET students' attitudes and reflections regarding smoking. Individual semi-structured interviews with teachers and management at VET schools were conducted to explore the potential for and acceptability of smoking prevention and cessation interventions. Both focus groups and individual interviews were audio-recorded and transcribed fully anonymised with codes assigned to each participant. Field notes and interview transcripts were analysed based on Malterud's systematic text condensation [41] to explore and identify recurring themes. The quotes were translated from Danish to English by the authors, adjusted for readability, and approved by co-authors. More in-depth analyses and more comprehensive findings are published as a separate need-assessment paper (forthcoming). We also used qualitative research methods in the feasibility studies of the identified intervention components (described under 'Overview of the development process', BCW step 8). See Table 1 for the qualitative data collected at VET schools.

**Table 1.** Data from fieldwork at VET schools.

| | Observations | Student Focus Groups (Number of Students) | School Staff Interviews | Workshops |
|---|---|---|---|---|
| *BCW stage 1:* | | | | |
| School 1 | 5 days | 1 (7) | 3 | |
| School 2 | 4 days | 1 (7) | 1 | |
| School 3 | 4 days | 1 (4) | 1 | |
| School 4 | 3 days | 1 (2) | 1 | |
| School 5–6 | | | | 4 [a] |
| School 7–9 | | | | 1 [b] |
| *Feasibility studies:* | | | | |
| School 10 | 1 day | 1 (7) | 3 | |
| School 11 | 1 day | 1 (8) | 3 | |
| School 12 | | 2 (10) | | |
| School 13 | | | | 1 [c] |

[a] Student workshops. [b] Stakeholder workshop with the participation of one school manager, two teachers, and one student from three VET schools. [c] Evaluation workshop: observation of the workshop and an interview with two of the workshop facilitators. The feasibility studies are described below (BCW step 8).

### 2.2.2. Workshops with Students

Further, we held four workshops with students from two other VET schools not mentioned above, offering education with TCT subject areas. Four student workshops with a total of approximately two hundred students from the basic programme at two VET schools were held in 2017 and facilitated by three consultants from Danish Committee for Health Education. The purpose of the student workshops was to gather knowledge about the students' perspectives on smoking and proposals for potential interventions to prevent and reduce smoking among the target group. In each workshop, the students, together with the facilitators, developed ideas for specific smoking initiatives with VET students

as a target group. The ideas were based on the students' own experiences with smoking and their perspectives on themes about social relations, school life, life in general, and tobacco legislation. The themes were initially presented with a short oral presentation. Since VET students may have resistance to smoking cessation and prevention efforts, it was described in the workshop introduction that the purpose of the workshop was to gain access to their experiences, attitudes, and ideas about how society, and in particular the school, could support non-smoking behaviour in everyday life. The workshop was structured in phases with brainstorming sessions and phases with short, intense sessions of group work with 6–8 participants in each group. In the final phase, the students had to select realistic ideas for smoking interventions. The results from the workshops have been published in a Danish language report with two authors (SA, CP) of this paper as co-authors [42].

### 2.2.3. Workshop with Stakeholders

A four-hour stakeholder workshop was held in June 2017. The workshop was organised in cooperation with the Centre for Prevention in Practice (CPP), Local Government Denmark, who also hosted and facilitated it as well as assisted in inviting representatives from VET schools and Danish municipalities who did not take part in this study otherwise. One school manager and two teachers from three VET schools were represented in addition to a representative from Danish VET schools' student organisation, representatives from six Danish municipalities, and three researchers (SA, JLV, and GSJ) (a total of sixteen participants). The purpose of the workshop was to discuss and share knowledge about smoking-related interventions at VET schools and to gather input and proposals to the development and planning of intervention components. We presented the findings from the COM-B analysis and the literature search as well as the overall ideas as an introduction to group-based discussions. The proposals and main conclusions from the discussions were written on a whiteboard in the room and noted on a computer for subsequent use in the development process. The workshop was not audio recorded. One researcher wrote notes during and after the workshop, and the facilitators noted key points made by the participants. These were used as supplementary data to help identify potential interventions and strategies.

### 2.2.4. Literature

We performed literature searches to inform several stages of the development process. The literature search in stage 1 ('Understand the behaviour') was performed to understand determinants for smoking behaviour among young people. The literature search for stage 2 and 3 ('Identify intervention options' and 'Identify content and implementation options') was conducted to give an overview of the intervention options and effects of smoking prevention and cessation interventions for young people. We used the databases PubMed and PsycINFO, and we created search blocks and defined relevant search terms. For each database, text word search terms, medical subject headings (PubMed), or thesaurus terms (PsycINFO) were used related to young people, smoking and interventions. Examples for PsycINFO: young adult*, youth*, student* and cigarette smok* and intervention, therapy, evaluat*. Moreover, we hand-searched for Cochrane reviews, Campbell collaboration reviews, and NICE reports. We complemented them with a literature search of existing Danish reports to gain information on the target group and the evaluation of existing tobacco-related interventions, including barriers, facilitators, and recommendations for implementing interventions. We used a snowballing method searching for grey literature through resources such as the Danish Center for Social Science Research and the Danish Cancer Society. A similar search was performed on Google and Google Scholar using search terms such as smoking, VET, young people, intervention, and similar terms in Danish.

## 3. Overview of the Development Process (BCW Steps)

In the following, we describe each BCW step. An overview of the data sources used in the different stages and steps is presented in Figure 1.

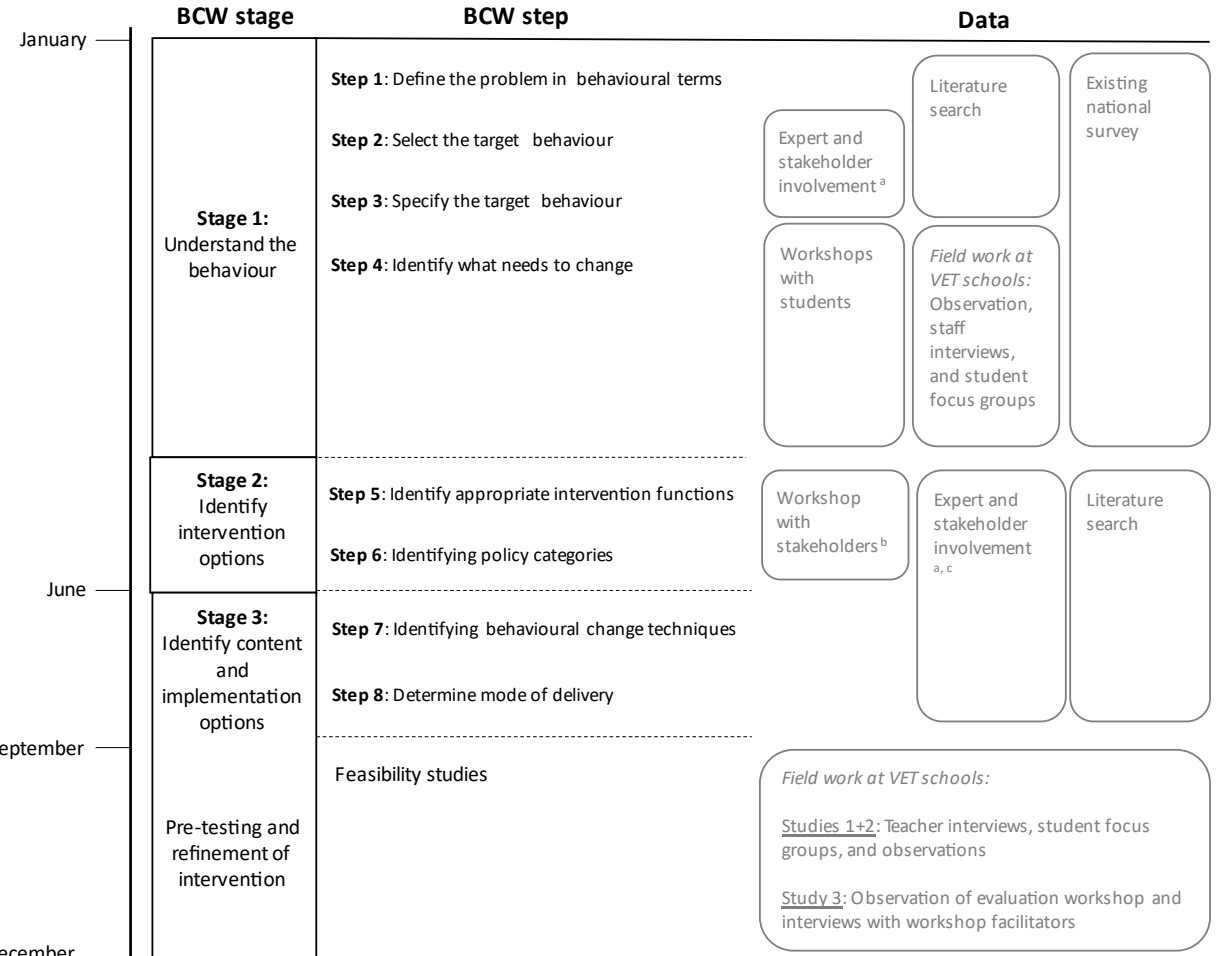

**Figure 1.** Overview of data sources for the stages and steps of the Behaviour Change Wheel (BCW) in the design of the intervention. [a] Representatives from the Danish Cancer Society, Centre for Prevention in Practice and a VET teacher/consultant. [b] Representatives from municipalities, VET schools and the student organization. [c] Representatives from the Danish Lung Foundation, the Danish Quitline and Copenhagen Municipality.

### 3.1. Step 1: Define the Problem

The first step was to identify and define the problem in behavioural terms. This was pre-specified by the members of the research group in the grant application and based on the national surveys previously conducted among VET school students [9] and principals [18].

### 3.2. Steps 2 and 3: Select and Specify the Target Behaviour

For the selection and specification of the target behaviour, we involved stakeholders and used knowledge from prior research. In the grant application, it was outlined that VET schools would be the context to target in the intervention.

### 3.3. Step 4: Identify What Needs to Change

In this step, we identified what needed to change by performing a COM-B analysis. We used the COM-B constructs to triangulate insights from the various data sources and

map barriers to reducing VET students' smoking. The definitions of each of the components of the COM-B are as follows [25]:

- Psychological capability: knowledge or psychological skills, strength or stamina to engage in the necessary mental processes;
- Physical capability: physical skills, strength or stamina;
- Physical opportunity: opportunity afforded by the environment involving time, resources, locations, cues, physical 'affordability';
- Social opportunity: opportunity afforded by interpersonal influences, social cues, and cultural norms that influence the way we think about things;
- Reflective motivation: reflective processes are cognitive processes such as goals and explicit attitudes and beliefs;
- Automatic motivation: automatic processes occur spontaneously without conscious control and are often based on affective associations related to seeking pleasure or avoiding displeasure.

### 3.4. Steps 5 and 6: Identify Intervention Functions and Policy Categories

Intervention functions are potential strategies to address deficits in one or more of the COM-B components (e.g., incentivisation could be used to address a lack of automatic motivation) [24]. We selected the most suitable and likely effective intervention functions based on the analysis of the COM-B components and TDF domains. How each of the intervention functions could be supported was determined by selecting relevant policy categories (e.g., regulation, environmental/social planning, communication/marketing). The intervention functions and policy categories were determined following iterative discussions between the authors and based on the workshops with students and stakeholders.

### 3.5. Step 7: Identify Content (Behaviour Change Techniques)

In this step, we identified specific behaviour change techniques (BCTs) under each broad intervention function category. A behaviour change technique is defined as the active component of an intervention that brings about a change in behaviour [25]. The BCW describes how each of the 93 BCTs is linked to the intervention functions. From the BCT taxonomy (BCTTv1) we mapped the specific BCTs most practicable and with promising effects on changing smoking behaviour in VET students.

### 3.6. Step 8: Identify Implementation Options (Mode of Delivery)

In the final step, we determined the mode of delivery of the intervention. We reviewed intervention options applicable to VET schools based on a literature search and included perspectives from stakeholders and feedback from VET students and staff. This led to the identification of the actual intervention activities in the Focus intervention. As part of step 8, we conducted three feasibility studies (September 2017–December 2017) to test the acceptability, practicability, and potential unintended effects of the identified intervention components, and further refine each intervention component prior to the final intervention. Methods are summarized below and in Table 1.

- Study 1: We pre-tested the class-based intervention components at two VET schools in the Capital Region of Denmark. We collected feedback through two focus group interviews with a total of fifteen students (age range 15–19 years; 65% women) and six semi-structured interviews with teachers, by discussing their experiences with the tested components. Moreover, we conducted two days of observations to observe how the intervention components worked in practice.
- Study 2: Smoking cessation support delivered by the national Quitline was discussed among students from a third VET school in the Capital Region of Denmark. We held two focus group interviews with a total of ten students (age range 16–18 years), which focused on students' acceptability of the Quitline approach.
- Study 3: A comprehensive school tobacco policy 'smoke-free-school-day' was implemented at a large VET school in the Region of Southern Denmark with support

from the Danish Cancer Society. Feedback on process was achieved through an evaluation workshop at the VET school, which was facilitated by the Danish Cancer Society. We observed the workshop and interviewed two facilitators from the Danish Cancer Society, who assisted with the implementation of the policy and facilitated the workshop.

## 4. Results

Below we present the outcome of the eight steps of the BCW framework.

### 4.1. Step 1: Identification of the Problem

The behavioural problem to be addressed was identified as high levels of daily smoking among VET students, and that interventions were needed to reduce smoking (as described in the introduction).

### 4.2. Steps 2 and 3: Specification of the Target Behaviour

Based on the findings from prior research and stakeholder involvement, we decided that interventions were desirable in the school settings to reach as many students as possible and to alter the school environmental conditions (as described in the introduction). In these steps, an important consideration in the research group was whether smoking cessation should be a specific target behaviour. We selected the whole student population as a target group because the social environment at school is influenced by social interactions which are centred on smoking. We identified two target behaviours: (i) reducing daily cigarette consumption, and (ii) preventing initiation of or escalation to daily smoking. We found that reduction of daily cigarette consumption would be a better behavioural objective that would be more realistic and reasonably achievable than smoking cessation per se. This decision was substantiated by evidence suggesting that individuals who reduce their number of daily cigarettes are more likely to attempt and ultimately complete smoking cessation later [43].

### 4.3. Step 4: COM-B Analysis: Identification of What Needs to Change

Using the COM-B model in our analysis, we sought to understand enablers and barriers to reducing smoking among students in VET schools. These factors are summarized in Table 2 and elaborated in the following.

**Table 2.** Overview of COM-B findings, intervention functions, and behaviour change techniques that the intervention addressed.

| COM-B | TDF | What Needs to Change | Intervention Functions (Policy Category) | BCW Techniques (BCW no.) | Intervention Elements |
|---|---|---|---|---|---|
| Psychological Capability | Knowledge | Students need to know how to receive smoking cessation support, and what are the benefits of the counselling | Education (C) | Instruction on how to perform a behaviour (4.1) | Information about the national Quitline in the edutainment session, and on posters. |
| | Knowledge | Students need to know about the impact of nicotine dependence | Education (C) | Information about antecedents (4.2), Information about health consequences (5.1) | Information in the edutainment session and the teaching material. |
| | Behaviour regulation | Students need to improve their self-regulatory ability and monitor their effort | Enablement Education Persuasion (C, E, S) | Biofeedback (2.6), Social support (unspecified) (3.1), Behavioural practice/rehearsal (8.1) | Class-based competition with carbon monoxide breath readings. |

| COM-B | TDF | What Needs to Change | Intervention Functions (Policy Category) | BCW Techniques (BCW no.) | Intervention Elements |
|---|---|---|---|---|---|
| Physical Capability | Skills | Students with nicotine dependence/craving need to be physically capable of not smoking | Education Enablement (S) | Instruction on how to perform a behaviour (4.1) Behaviour substitution (8.2) | Advice by/talks with staff who have been trained on the staff course. The national Quitline service adapted to VET students. |
| Physical Opportunity | Environmental context and resources | Students need to have access to smoking cessation services and support at school | Enablement Environmental restructuring (E) | Instruction on how to perform a behaviour (4.1) | Information in the edutainment session and posters at school on how to contact the national Quitline. Staff trained in having dialogue with students. |
| | Environmental context and resources | Schools need to make smoking more difficult on school premise and close to the school | Restriction (R) | Restructuring the social environment (12.2.) | Implementation of a comprehensive school tobacco policy, where students, staff, and visitors are not allowed to smoke during school. |
| Social Opportunity | Social influences | Students need a supportive social environment at school and to alter the perception that smoking is the social norm, or not feel obligated to adhere to the perceived norm. | Enablement Environmental restructuring Modelling (C, E) | Social support (unspecified) (3.1.), Demonstration of the behaviour (6.1), Social comparison (6.2), Information about others' approval (6.3), Habit transformation (8.3) | School tobacco policy and quit-and-win competition: experiencing others who do not want to initiate smoking or want to quit. The teaching material: correcting misperceptions of overestimation of smoking prevalence; student involvement in class ethos and creating social activities in breaks. |
| Reflective motivation | Social role and identity | Students need to minimise their perception of smoking as part of their social engagement at school and their identity as being young | Education Persuasion (C, E) | Identification of self as a role model (13.1), Framing/reframing (13.2) | The teaching material: for example, discussing the responsibility of being a role model for younger students; how smoking is influenced by, e.g., family, friends, school, legislation, the tobacco industry |
| | Beliefs about consequences | Students need to have fewer positive beliefs about the psychological and social benefits of smoking | Education Persuasion (C, E) | Information about social and environmental consequences (5.3) Information about emotional consequences (5.6) | The teaching material: addressing beliefs about benefits of smoking. |
| | Beliefs about capabilities | Students need to correct their belief that they can quit at any time without assistance | Education Persuasion (C, E) | Framing/reframing (13.2) | Edutainment session and teaching material: smoking reframed as an addiction, not a choice. |

**Table 2.** *Cont.*

| COM-B | TDF | What Needs to Change | Intervention Functions (Policy Category) | BCW Techniques (BCW no.) | Intervention Elements |
|---|---|---|---|---|---|
| Reflective motivation | Intentions | Students need to increase awareness of the negative effect of smoking on their body and be encouraged to reduce smoking | Incentivisation (C, S) | Biofeedback (2.6) | Monitor the students carbon monoxide levels twice. |
| | Goals | Students need to have a vision of what they achieve by reducing smoking | Education Training (S) | Goal setting (behaviour) (1.1.), Goal setting (outcome) (1.3.) | By participating in the competition, the students want to reduce own level of carbon monoxide. |
| Automatic motivation | Reinforcement | Students need to change their perception of smoking as a habit in their daily school life | Environmental restructuring Restriction (E, R) | Restructuring the physical environment (12.1), Restructuring the social environment (12.2), Behavioural practice/rehearsal (8.1), Habit formation (8.3) | School tobacco policy: encouraging habit formation, changing the habit during breaks, commitment not to smoke during school, creating fear of being caught for violating the policy. |
| | Reinforcement | Students need to have a tangible encouragement to reduce smoking | Incentivisation (S) | Incentive (outcome) (10.8), reward (outcome) (10.10) | Competition and the prize: rehearsal, creating expectation of rewards, celebrating wins. |
| | Emotion | Some students experience that they need smoking to cope with stressful situations. | Education Persuasion (E) | Social support (emotional) (3.3.), Information about emotional consequences (5.6), Reduce negative emotions (11.2), Framing/reframing (13.2) | Support from staff. The teaching material: discussing beliefs about psychological benefits of smoking and how to tackle stress. |

Abbreviations: COM-B: capability, opportunity, motivation—behaviour; TDF: Theoretical Domains Framework. Policy categories: G: guidelines, C: communication/marketing, E: environmental/social planning, F: fiscal measures, R: regulation, S: service provision.

### 4.3.1. Psychological Capability

The focus group participants in stage 1 identified some main areas of the students' psychological capabilities, which would be possible to change through an intervention. Although the students knew that smoking constitutes a severe health risk in the long term, they did not express knowledge about the immediate harms of smoking on, e.g., the cardiovascular system and the brain. Nor did the students express awareness about nicotine dependence and its influence on their ability to quit smoking. Rather, the students described smoking as a "*harmless*" act and a natural part of being young. Finally, and importantly, the students lacked knowledge about access to smoking cessation support.

### 4.3.2. Physical Capability

Survey responses revealed that 29% of VET students smoked daily, and 57% of the students who smoked daily were physically dependent on smoking and experienced symptoms of physical nicotine dependence, as measured by an item from the Fagerström test for nicotine dependence (responded 'within 5 min' or '6 to 30 min' to 'How soon after you wake up do you smoke your first cigarette?') [7]. Research has shown that a large proportion of young people who smoke daily are addicted to nicotine but unaware of their level of nicotine dependence and often attempt to quit without any support [14,44].

Particularly, people from low socioeconomic backgrounds more often make quit attempts without seeking cessation counselling, despite being highly dependent [45,46].

### 4.3.3. Physical Opportunity

School tobacco policies have generally been liberal in Danish VET schools [19]. At the time of this study, smoking was allowed in designated outdoor smoking areas, unless the school shared premises with a high school; then smoking was not allowed on school premises [19]. A teacher on a VET school sharing outdoor premises with a high school explained how the smoking practice was on his school:

> "*Due to the legislation regarding young people under the age of 18, smoking has been relocated [from the school premises] to public roads. Smoking is not allowed on school premises. But luckily you can still go outside to smoke. But besides that, I don't have any impression that anything is being done here.*"

Data from a national survey among principals showed that all VET schools had a tobacco policy, but with substantial variation in its comprehensiveness and enforcement [18]. Moreover, only a third of VET schools offered some kind of smoking cessation support (permanent or periodical), and there was an extensive lack of information about smoking cessation assistance [19]. The Danish Cancer Society has been involved in educating school staff as smoking cessation counsellors with the aim of offering local smoking cessation courses. However, experiences with the courses have been that the student enrolment was low [19].

### 4.3.4. Social Opportunity

In the workshops, students reported that taking up smoking when starting at the vocational school was an easy way to socialize:

> "*It is frustrating that it is easy to socialize if you smoke. Enrolling in a new school, the non-smokers stand in each corner, and don't know where to go. It is much easier to get new friends, if you are a smoker, when you enrol in a new school.*"

Hence, the students' participation in smoking was closely linked to their efforts to make new social relationships but also to maintain existing social relations. According to some students in both workshops and focus groups, the designated smoking areas were the cosiest places on the school ground. A non-smoker explained how she also joined the smoking area: "*Sometimes you just tag along for the fun of it*". A student also described how he experienced smoking as a social pressure:

> "*There is social pressure, i.e., if you have a dominant group in class that you look up to, it affects whether you smoke or not, because you want to be part of that group.*"

We identified a general lack of positive non-smoking role models both in schools and outside school. Several participants in the focus groups also explained how they expected people to smoke at VET schools and in vocational professions.

### 4.3.5. Reflective Motivation

A key factor identified was that students understood smoking as an integral part of being young, and they described smoking as a "*cool*" behaviour. Students believed that "*smoking is my own choice*", and they expressed the attitude that they were now part of an adult education system, no longer in compulsory school, which granted them the freedom to choose for themselves. These findings highlight that social role and identity are important factors to target in the intervention. Smoking was also used as a method to withdraw from the social life at school. Two students explained how smoking a cigarette enabled some relaxing time alone: "*Then it feels good to just be alone*" and "*Yes, then you can sit by yourself*". In general, the students who smoked conveyed cessation as only a matter of will power and expressed the belief that it would be easy to quit later in life. Nevertheless, they recognised the impact of perceived stress on their ability to quit smoking.

### 4.3.6. Automatic Motivation

Students interviewed described how the habits during school were entangled with smoking, i.e., breaks were understood as synonymous with smoking or smoking was described as a habit that coincided with breaks or boredom: "*We just do it [smoke] without thinking about it. It's like drinking coffee*". Other students gave accounts that suggested nicotine dependence. In other words, some students explained how they associated smoking with relief from negative emotions: "*[you smoke] If you're in a bad mood*", and "*If you're upset or something*". Particularly, smoking was reported to relieve stress and anger: "*If you've been in a fit of rage, you just need to de-stress a bit*". It is a widespread view among young people who smoke regularly that smoking decreases their negative emotions [47]. As such, both automatic and reflective motivation were identified as important determinants for whether VET students would reduce smoking during school.

### 4.4. Steps 5 and 6: Identification of Intervention Functions and Policy Categories

After identifying the COM-B components relevant to the targeted behaviours and based on the stakeholder workshop and consultations with experts, we selected education, persuasion, incentivisation, restriction, environmental restructuring, modelling, and enablement as appropriate intervention functions relevant to reducing smoking at VET schools. The intervention functions were subsequently mapped to policy categories (see Table 2). For example, 'environmental/social planning' was chosen to support the delivery of the intervention function 'environmental restructuring' (to bring about change in the social opportunity to smoke at school and provide access to smoking cessation support), 'regulation' was chosen to support the delivery of 'restriction' (to handle the physical and social opportunity to smoke at school), 'communication/marketing' was chosen to support the delivery of 'education' (to increase knowledge and understanding of why the environment and nicotine dependence are important factors in smoking), and 'service provision' was chosen to support the delivery of 'enablement' (to have easy access to smoking cessation support in school) and 'incentivisation' (to create expectation of reward by lowering own level of carbon monoxide).

### 4.5. Step 7: Identification of Behaviour Change Techniques (BCTs)

An overview of the selected BCTs, with links to what is needed to change, and intervention functions, are shown in Table 2. The BCTs identified for the intervention were those which we considered the most promising to elicit less smoking at Danish VET schools. In total, we identified 23 BCTs to be potentially effective and appropriate for the intervention. Others' approval, social support, and restructuring the physical and social environment were identified as some of the appropriate BCTs to handle the opportunity to smoke and influence the automatic motivation. 'Instruction on how to perform behaviour' and 'feedback in behaviour' were identified as some of the appropriate BCTs to increase the capability to change smoking behaviour, along with the information about social and environmental consequences, reframing, and identification of students themselves as role models as some of the appropriate BCTs to influence reflective motivation.

### 4.6. Step 8: Identification of the Mode of Delivery

Based on the prior steps, together with interventions identified in the literature and discussions with relevant stakeholders (e.g., the Danish Cancer Society and CPP), we devised a list of preliminary ideas and selected potential intervention components. This included elements at the school, class, and individual levels within the school settings, as outlined below.

### 4.6.1. School Environmental Component: School Tobacco Policy

We chose a comprehensive school tobacco policy defined as smoke-free school hours, where students, staff, and guests are not allowed to smoke during school hours. We were worried that the smoking ban would counteract students' intrinsic motivation for reducing

smoking. The core hypothesis of SDT is that the process of internalization is supported by satisfaction of the psychological needs for autonomy, relatedness, and competence [28,29]. To support autonomy and allowing the students to feel being in control, we emphasized that school staff should relate to students in an empathic, non-judgemental manner even if they violate the school tobacco policy. To support the need for relatedness, we tested two class-based intervention components: 'walk and talks' that included a focus on the class culture, the social curriculum and students' perspectives on smoking in their everyday life and a 'quit and win competition' with a prize for the whole class. Finally, we decided to include an intervention component to support or build competences for quitting smoking. Students who indicate intention to quit smoking or needed support during school hours could use the national Quitline (named 'Stoplinien' in Danish). These intervention components are described below.

### 4.6.2. Class-Based Component: Walk and Talks

This intervention component comprised weekly class-based walk-and-talks providing students with dialogue cards with a set of questions facilitating conversations between the students on topics related to class culture, smoking habits, stress, etc. The questions were formulated by the researchers based on materials from the Danish Lung Foundation and the Danish Cancer Society. This format of delivery was chosen based on discussion with a current VET teacher and former researcher with in-depth knowledge on and expertise with Danish VET schools. The VET teacher and her colleagues had good experiences with walking with the students during the class, which provided the basis for a dialogue due to the fresh air and energy spent and an opportunity to talk more informally and non-academically. Moreover, group-based intervention strategies have the benefit that participants can provide each other with support and encouragement, as well as strategies, e.g., resisting social pressures [22].

### 4.6.3. Class-Based Component: Quit and Win Competition Based on Measurements of Carbon Monoxide Levels

We further chose a class-based competition based on carbon monoxide (CO) measurement as the mode of delivery. This choice was informed by suggestions from the students themselves [42]. The students' CO levels were orally measured twice: at baseline and 10-week follow-up. The class with the largest overall reduction or maintenance of CO levels won a prize comprising a social activity, i.e., a bowling trip. By measuring individual CO, we also aimed to raise attention and interest in the school tobacco policy and, moreover, to provide them with biofeedback, which was one of the identified BCTs. The competition part further aimed to encourage students to either reduce smoking or avoid initiating smoking by creating an expectation of rewards. The underlying idea was to motivate the students, without focusing on the hazards of smoking, rather the motivation was an immediate, tangible award. Further, the element of competition and the desire to win the prize were thought to strengthen the sense of unity and social relations between students in the class while potentially spurring non-smoking students to support their smoking peers in quitting or reducing smoking.

### 4.6.4. Individual-Based Component: Access to Smoking Cessation Support

We chose to deliver information about the Danish national Quitline ('Stoplinien'). Stoplinien offers reactive telephone cessation counselling to those who wish to stop using tobacco products. We choose this delivery format due to its anonymous and confidential nature [48], and because telephone counselling has been presented in previous research as a promising format [49].

## 5. Feasibility Testing and Subsequent Intervention Refinement

Findings of the feasibility testing of each intervention component are summarized below.

*5.1. Smoke-Free School Tobacco Policy*

Knowledge from the evaluation workshop and interview with the workshop facilitators revealed that the school adhered to the smoke-free school policy by formal consensus on the policy, but the staff experienced misunderstandings regarding the specific time span the ban applied. A workshop facilitator explained:

*"... they interpreted the smoke-free school time differently, i.e., when does the school open (is that when you meet in the morning or do we uphold official hours?) and likewise, when a lecture ends, is the school closed then? It may be [closed] even if you're doing group work [in the evening]."*

Therefore, we refined this intervention component to include an implementation manual intended for school management. In the manual, we emphasised the significance of communicating clear rules, as specified by a fixed time span (e.g., 7 a.m.–5 p.m.), and that the school management has the responsibility to communicate the rules to the students, staff, and visitors.

Another subtheme from the interview with the facilitators from the Danish Cancer Society was that teachers and managers reported a dilemma regarding the school staff being subjected to smoke-free working hours while they were given a new role in enforcing the smoking rules. In other words, some teachers, who smoked themselves, felt resistant or ambivalent towards the new policy and thus found enforcing it difficult. Moreover, findings from the interview underlined that the implementation of the 'smoke-free-school-hour' policy requires regular support for students who find it difficult to avoid smoking during school hours, and that regular support should preferably be delivered by the school staff who are part of everyday life at the school. To accommodate this, we included a two-day staff course on a dialogue about smoking aimed to support professionals who work with young people. The course was developed by the Danish Cancer Society. On the first day of the course, the participants were trained in tools for supporting students based on counselling techniques comparable to the "5 A" method (i.e., Ask, Advise, Assess, Assist, Arrange) that has shown promising results, e.g., in the American high school settings [50]. On the second day of the course, the participants were given the opportunity to discuss and plan motivational and social activities at their school to further support the implementation of the smoke-free school policy.

Participants at the stakeholder workshop described that staff at VET schools commonly experience a lack of time, competing priorities, and a lack of interest in tobacco projects. Therefore, we developed materials and means to support the intervention components locally at each school. In addition to the implementation manual, this included a booklet covering the teaching material to be presented and discussed at meetings between the teachers and researchers, meetings with school management followed up with e-mails or phone calls aimed to discuss challenges and solutions.

Based on the experiences from the feasibility study, we decided to implement the smoke-free school policy after the summer holiday, at the beginning of a new semester. About three months before the semester started, the Danish Cancer Society held the staff course for three to four teachers (or other relevant staff) at each school. However, the stakeholders from the Danish Cancer Society also emphasized that: *"In the first months after introducing it [the policy] you need to do something to increase visibility about the new rules ..."*. Therefore, to further facilitate and highlight the school policy, we included an additional intervention component, namely an edutainment session (a lecture that both educates and entertains). The edutainment session was to take place at the beginning of the intervention period to boost the implementation of the school tobacco policy. The edutainment session was held by an external professional actor who had already developed a concept for younger people in compulsory school. The actor adjusted the content to VET students. The edutainment session was a one-time event for students to promote their knowledge about nicotine dependency, the consequences of smoking, the individual risk of illness, and common misperceptions about smoking delivered with a combination of

information and entertainment. This involved highlighting the influence of the tobacco industry, and the specific aim was to reframe smoking as an addiction and not a free choice.

### 5.2. Class-Based Component: Walk and Talks

The feasibility study underlined the relevance and appropriateness of the topics provided for discussion in the walk and talks but prompted us to reconsider whether the mode of delivery was appropriate. The students were not able to focus on the assigned topics when they walked around in unsupervised groups, and they discussed the topics for a shorter time than anticipated. The teachers' introduction to the activity was important in terms of how it was perceived by the students. Students discussed the topics in-depth and more intensively when facilitated in person by a teacher or a researcher, as said by a student in one focus group:

> "It didn't go very well when we walked around by ourselves, because then we'd just talk, like half a minute per question, and then we went back up [to the classroom]. But when for example you [a researcher] went along, then we would talk a lot about it, and more and more questions would come up, and then we would actually talk more about it [the assigned topic]."

An unintentional result of the walk and talk as the mode of delivery was that many students used it as an occasion to smoke. As another student said:

> "I find it really ironic, that when we are going to do a non-smoking thing, then why does half of the class light up cigarettes, it's because no one is checking up on us. [ . . . ] So, I think it should be more structured."

The students suggested that such discussions would work better, and be more structured, if conducted as a regular lesson in the classroom and led by a teacher. Such setup would more likely include each students' opinions, and everyone would have the chance to say something as the discussions would be moderated by a teacher. Therefore, based on these inputs we decided to develop a compendium with material for eight classroom sessions. The material still included subject cards with a set of questions as well as subjects for discussions. To target social support and habit transformation, the first session was concerned with involvement of the students in class ethos and creating social activities in breaks.

### 5.3. Class-Based Component: Quit and Win Competition

The feasibility testing showed that students overall appreciated the competition as well as the prize. They highlighted the fact that the prize was not school-related and enjoyable for the class as a group. Interestingly, some students even expressed that the competition was the only motivation to participate in the project underlining the importance of including incentivisation as an intervention function. As one student said:

> "If someone told me 'you won't get anything if you do this', then I wouldn't do anything. Because today, you have to get something to do something. So, if someone told me that 'you won't get anything if you do this [stop/reduce smoking]', then there wouldn't be any benefits to gain from it."

Another student who was also a smoker emphasized the importance of measuring the CO level:

> "I mean even though there wasn't a competition about a bowling trip, I still think it could be fun to see if your CO levels went down. And to sort of see if there was any change. That would be a challenge to yourself, to see how much you could get the number down."

Students described the competitive intervention component in positive terms. Several students expressed appreciation of how the project started out with the measurements and how it contributed to awareness raising and making the intervention personally relevant. One student, who was a smoker, said:

*"That you started out measuring the CO levels ( . . . ) it was very good that you started with that. Because then all the smokers could start out seeing, that it really has an effect on us, that we're smoking."*

As such, the measurement of CO levels appeared to motivate the students both in itself and as a competitive element. Nevertheless, some non-smoking students thought the competition was too focused on smokers, and they felt excluded as they could not contribute to the competition, because they did not smoke:

*"Well, I think that it's a cool concept. That it's a bit of competition, you think you want to be better than the others or something silly like 'we're gonna take them down' (laughing), I think that's cool. But I can't do much because I don't smoke, so I can't really be that person that helps."*

Based on this feedback, we reassessed how to best communicate this intervention component underlining the importance that non-smoking students remain non-smokers while simultaneously supporting classmates who smoke in their efforts to reduce their consumption.

### 5.4. Individual-Based Component: Access to Smoking Cessation Support

The feasibility study showed that students' responsiveness to the smoking cessation support provided by the Quitline was overall rated positively. However, they emphasized that a positive relationship must be established between the person wanting to quit and the counsellor, while the enrolment for the smoking cessation support also must be discreet. According to some students, the possibility of joining in couples or groups could be advantageous. Nevertheless, the students expressed that smoking cessation support would only help if the students themselves were motivated to quit:

*"If I was determined to quit, I think it would be helpful. But it depends on yourself. If I had decided 'Okay, I need to quit now' then I believe it could help me, that it could give me that extra boost. And if you did it together with a group."*

Together with the manager of 'Stoplinien' and one cessation counsellor, we adapted the cessation support component to include counselling on how to refrain from smoking during school. Moreover, based on students' feedback and prior research [51], the Quitline service was refined to be flexible allowing students to participate in pairs and with multiple call-back sessions. Furthermore, we decided to deliver information about the service from Stoplinien through (i) a video session integrated in the above-mentioned edutainment session in which a smoking counsellor delivered information about the support, and the students were given the opportunity to anonymously write their telephone number on a note, which the actor delivered to the counsellor with the aim to proactively support the students by telephone, (ii) information posters placed on school premises, and (iii) information flyers delivered to the school staff who attended the course on the dialogue about smoking.

## 6. Final Intervention

To describe the final intervention, we followed the Template for Intervention Description and Replication (TIDieR) checklist [52], see Table 3.

**Table 3.** The TIDieR (template for intervention description and replication) checklist for the Focus intervention.

| Checklist Item | Item Description |
| --- | --- |
| Name | Focus: a school-based multicomponent intervention focusing on reducing smoking in the school environment for vocational education and training (VET). |
| Why | Danish VET schools have a high number of students who smoke. |

Table 3. *Cont.*

| Checklist Item | Item Description |
|---|---|
| What [a] | 1. School tobacco policy on smoke-free school hours, where students, staff, and visitors are not allowed to smoke during school, specified by a fixed time span (e.g., between 7 a.m. and 5 p.m.).<br>2. Course for school staff on short motivational counselling about smoking intended for professionals working with young people.<br>3. Edutainment session on tobacco products.<br>4. Classroom-based teaching according to themes about attitudes, beliefs, and social influence as well as wellbeing at school.<br>5. Class-based quit-and-win competition according to measurements of carbon monoxide (CO) levels.<br>6. Information about and easy access to smoking cessation support offered by the national Quitline. |
| Who provided | 1. School management, teachers, and student counsellors or other relevant staff as desired by the school.<br>2. One consultant and one psychologist from The Danish Cancer Society.<br>3. An actor.<br>4. Teachers.<br>5. Researchers.<br>6. Counsellors from the national Quitline. Posters were developed by the researchers. |
| Where | The intervention itself occurred in school during school hours. The staff course took place at the University of Southern Denmark. |
| When and how much | 1. The school tobacco policy was implemented during a semester from school start in August.<br>2. The two-day staff course was delivered a few months before. Training was held off-site for all intervention schools together.<br>3. Class lessons comprised eight sessions that could take place during the semester (e.g., over eight weeks)<br>4. The measurements of carbon monoxide levels occurred in August and October. The prize was delivered in November.<br>5. The edutainment session was delivered to students and their teachers and other relevant staff in the first weeks of the semester.<br>6. The Quitline offered three telephone sessions. |

[a] See Supplementary Materials (Table S1) for a description of materials and procedures for each component.

## 7. Discussion

Knowledge of promising interventions to assist young people in quitting and prevent smoking initiation and escalation is highly demanded. This paper describes the systematic theory- and evidence-based development of a smoking-prevention-and-reduction multi-component intervention for VET schools in Denmark. The comprehensive development process based on the BCW framework led to an intervention that addresses less smoking at VET schools simultaneously at both school structural level, class level, and individual level. We chose a comprehensive school tobacco policy (smoke-free school hours) as the mode of delivery, along with class-based educational and motivational activities, and improved access to smoking cessation support. Our choice reflects a complex interaction of influences on student smoking in VET schools.

Comprehensive school tobacco policies are recognised as promising in promoting a non-smoking behaviour at school. Kuipers et al. investigated the association between school tobacco policies and students' smoking, using data on students aged 14–17 from six European countries. They found that more restrictive policies were associated with less smoking in schools, though not associated with overall daily smoking rates [53]. Similarly, another study found that higher levels of school tobacco policies, as perceived by the students, were associated with less smoking on school premises [54]. This suggests that school policies may reduce smoking during school, but are less successful in reducing the overall cigarette use [55]. In addition, a realistic review demonstrated that school smoking policies may trigger unintended cognitive and behavioural responses [56]. The barriers identified in the review include students finding alternative places to smoke or developing

counterproductive views about the purpose of the policy (e.g., rules exist only to protect the reputational standing of the school). Moreover, lack of consistent enforcement and staff who undermine adequate implementation have been identified as barriers for implementation [56,57]. These findings point to the importance of supplementing restrictive school tobacco policies with motivational, enjoyable, and supporting elements. This complies with previous publications which show that more positive outcomes are obtained for programmes adopting a 'whole-school' approach, where a balance of both universal and targeted approaches has been recommended [58,59]. This aligns with our finding that cessation support should be included. Systematic reviews on smoking cessation interventions for young people have reported limited evidence on successful interventions [22,60] but support potential benefits of quit-and-win contests [60] and smoking cessation services [12,61], which we have considered in the development of Focus.

A strength of our study is that we report changes in the intervention content during the development, as recommended in the guidance for the reporting of intervention development (the GUIDED checklist) [62]. Our study demonstrates the necessity of including feasibility testing as part of developing an intervention. We included initial steps to examine if the developed intervention components were feasible in the VET school context and acceptable for VET students. This helped us to identify key uncertainties in delivery and to adjust and refine the intervention components. Another strength of this study includes the fact that we followed the recommended stages of BCW, assessed the empirical and theoretical literature and included several data sources: national surveys, fieldwork and workshops and continuous engagement of key stakeholders. This enhanced identification of promising intervention components. Moreover, our cooperation with the Danish Lung Foundation and the Danish Cancer Society in developing the teaching materials enables the possibility that these organisations could ensure easy access to the materials for the schools. Moreover, we adhered to TIDieR guidance in its description to enable replication by others. An important contribution of our article is the detailed description of how we developed our intervention in practice.

However, we also experienced limitations. First, we included only changes in students' behaviour, and not the changes in the smoking behaviour of school staff who are role models for the students and play an important role in supporting the intervention. Although the school tobacco policy and smoking cessation offer include the school staff, they are not targeted by any motivational efforts for smoking cessation. It might be beneficial to focus on how to reduce smoking among school staff, as our qualitative fieldwork showed that the behaviour of some teachers legitimized smoking at the school. The limited number of school staff included in the initial fieldwork may have left this underexplored. Had we emphasized staff perspectives in the data collection, or used implementation-based frameworks, or partnership approaches where school staff and students had co-designed the intervention with us, we might have had a better starting point to consider potential issues about implementation [63]. Second, limitations were found in our use of the BCTs. The environmental context and resources were identified as important enablers and barriers to reducing smoking at VET schools. Still, there is a lack of BCTs focusing on how the surrounding environment has positive or negative effects on individual behaviour. In this regard, Intervention Mapping may have helped to address environmental factors [64]. The BCTTv1 system requires some adaptation to incorporate and specify environmental and interpersonal relational aspects of interventions that use approaches that incorporate environmental influences. Second, the BCT taxonomy is complex to use and requires training. Although members of the research team have some experience with using the BCW method, we may have overlooked some BCTs or misplaced some of the identified BCTs.

## 8. Future Directions

Future studies can learn from our development work and repeat stages in the development of interventions. The intervention development process described may be useful in developing other interventions that use approaches integrating cognitive, behavioural,

and environmental influences. To optimize intervention development and accumulate knowledge on this important task, we recommend that other intervention developers be transparent about how they develop their intervention. Moreover, the BCT coding can help future reviews and meta-analyses of interventions in identifying effective BCTs. Our findings underline the importance of feasibility studies to test preliminary assumptions on behaviour and uncover unintended consequences of intervening. For example, we a priori assumed that walk and talks with question cards would be an appealing and suitable method for stimulating students to reflect upon their smoking practice and reasons to smoke; however, the feasibility study showed, that the students preferred desk teaching for this discussion. Our study also showed that some students used the walk and talks to smoke, reflecting an unintended outcome of our intervention.

## 9. Conclusions

Using a systematic, theory- and evidence-based, and setting-informed process, we have developed and described a school-based intervention, targeting smoking reduction among youth in vocational school settings. The BCW framework was useful to identify potential active ingredients of the intervention components. We identified important barriers to reducing smoking among VET students, which included the smoking culture in VET schools, and the students' habits and beliefs related to the social and psychological effects of smoking. Several intervention components were identified, pre-tested, and modified. These included changing the social and physical environment of smoking by implementing a restrictive school tobacco policy, increasing capabilities to reduce smoking by offering better access to cessation support, and targeting motivation by providing a curriculum with teaching material and a quit-and-win competition. Environmental factors and social opportunities could prove essential when developing interventions for young people at schools with a high proportion of smoking.

**Supplementary Materials:** The following supporting information can be downloaded at: https://www.mdpi.com/article/10.3390/youth3020047/s1, Table S1: Materials and procedures.

**Author Contributions:** Conceptualization, S.A. and J.L.V.; methodology, S.A., T.H. and R.F.K.; formal analysis, J.L.V., D.D. and D.H.H.; investigation, J.L.V., S.A. and D.H.H.; data curation, G.S.J.; resources, M.P.J.; writing—original draft, S.A.; validation, R.F.K., T.H. and C.P.; Visualisation, M.P.J.; writing—review and editing, all authors; funding acquisition, S.A. All authors have read and agreed to the published version of the manuscript.

**Funding:** This research was funded by the Danish Cancer Society and TrygFonden (Denmark).

**Institutional Review Board Statement:** The study was conducted in accordance with the Declaration of Helsinki and approved by the Danish Data Protection Agency (record number: 17/12006) prior to collecting data. The National Committee on Health Research Ethics concluded that formal ethics approval was not required because no human biological material was sampled (record number: 20182000-83). There is no formal institution for ethical assessment and approval of register- and questionnaire-based population studies in Denmark.

**Informed Consent Statement:** Informed consent was obtained from all people involved in the interviews. Students and teachers present during participant observations and those participating in interviews were informed about the aim of the study, which was introduced as a study about smoking, health behavior, and well-being of students in VET schools. Furthermore, they were informed that all data was used for research purposes only, treated with confidence, and anonymized in publications.

**Data Availability Statement:** Not applicable.

**Acknowledgments:** The authors want to thank all students, teachers, and staff at the Danish VET schools for participating in observations, interviews and focus groups, and to anthropologist and special consultant Betina Bang Sørensen at NEXT Education Copenhagen (large school offering VET programmes in locations around the greater Copenhagen area) for assisting on the area of VET schools. Also, thanks to the Danish Cancer Society, the Copenhagen Municipality, Local Government Denmark and the Danish Lung Foundation. Finally, thanks to Sandra Nielsen for her contribution

during an internship at University of Southern Denmark as part of her study in master in Public Health at Göteborg University.

**Conflicts of Interest:** The authors declare that they have no competing interests.

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
