# Peer review of "Development and Feasibility Test of a Theory- and Evidence-Based Multicomponent Intervention to Reduce Student Smoking at Danish Vocational Schools"

_2673-995X, doi:10.3390/youth3020047_

Round 1

Reviewer 1 Report

The manuscript has several potentials although there are some issues that need to be addressed.

Here are my specific comments:

1) In the abstract you introduce the term, "an edutainment session". Can you add a couple of words to explain what it is as done in the main text?

2) In the introduction, where you talk about the alarming prevalence of smoking among students attending vocational education and training, a rather old reference from 2014 is used. A more recent reference can be appropriate to include here.

3) In the introduction as well, a more detailed description of the two theoretical frameworks: the behaviour change wheel and the self determination theory is needed along with some information on their empirical evidence.

4) In the method section, you only need to include information on how the theoretical frameworks have been used in the intervention.

5) Under the literature section on page 4, you talk about stages of the development process. Please include also the names of the stages for ease of reading.

6) In the Results section, under physical capability, you mention that ".... 57% of VET students who smoked daily were physically dependent on smoking....". Can you also include the proportion that smoked daily? This will provide some information regarding the extent of smoking in VET students.

7) Table 3 is a bit crowded and not easy to follow. Please restructure it.

8) Implications of the findings, with respect to further research, policy formulation, and practice (i.e., how your intervention can inform others) should be more explicit and ideally put under its own heading. 

9) The paper is generally well-written but there are some language and structural issues that need to be corrected. For example:
- there appears to be some problems with the numbering system for sources of data. Please check.
- under "Overview of the development process (BCW steps)", you write "Step 2 and 3", but it should be "Steps 2 and 3".
- On page 6, study 3, you have "Feedback on proces...." instead of "Feedback on process....".
- On page 15, in the second quotation, you have  the following sentence which does not read well: "But it obviously motivates more that there is the bowling trip that you can win". Please check.

Reviewer 2 Report

Introduction:

-It would be helpful to provide a brief discussion in the introduction section about school tobacco policies in Denmark. It was briefly mentioned later in the results that Danish VET schools are “liberal,” but it is unclear what this looks like.

Study setting:

- “VET lead to a VET qualification” – add s to the word lead

-“ the prescribed duration is a least three years.” – change a to at

-What is the reasoning why the authors only interviewed 1 school manager, 1 student advisor, and 4 teachers total in the four VET schools? This limits the in-depth understanding of the authors regarding the development of smoking prevention and cessation interventions. Consider explaining this in the limitation section.

-Are the VET schools that the researchers targeted for the interventions representative of the Danish population? It would be useful if the authors can provide a brief summary of the demographics of these schools and the reason why these schools were chosen
